# The Scaffold Protein PICK1 as a Target in Chronic Pain

**DOI:** 10.3390/cells11081255

**Published:** 2022-04-07

**Authors:** Andreas Toft Sørensen, Joscha Rombach, Ulrik Gether, Kenneth Lindegaard Madsen

**Affiliations:** Molecular Neuropharmacology and Genetics Laboratory, Department of Neuroscience, Faculty of Health and Medical Sciences, University of Copenhagen, DK-2200 Copenhagen, Denmark; andreass@sund.ku.dk (A.T.S.); joscharombach@sund.ku.dk (J.R.); gether@sund.ku.dk (U.G.)

**Keywords:** neuropathic pain, inflammatory pain, drug development, therapeutic peptides, maladaptive plasticity, calcium permeable AMPAR

## Abstract

Well-tolerated and effective drugs for treating chronic pain conditions are urgently needed. Most chronic pain patients are not effectively relieved from their pain and suffer from debilitating drug side effects. This has not only drastic negative consequences for the patients’ quality of life, but also constitute an enormous burden on society. It is therefore of great interest to explore new potent targets for effective pain treatment with fewer side effects and without addiction liability. A critical component of chronic pain conditions is central sensitization, which involves the reorganization and strengthening of synaptic transmission within nociceptive pathways. Such changes are considered as maladaptive and depend on changes in the surface expression and signaling of AMPA-type glutamate receptors (AMPARs). The PDZ-domain scaffold protein PICK1 binds the AMPARs and has been suggested to play a key role in these maladaptive changes. In the present paper, we review the regulation of AMPARs by PICK1 and its relation to pain pathology. Moreover, we highlight other pain-relevant PICK1 interactions, and we evaluate various compounds that target PICK1 and have been successfully tested in pain models. Finally, we evaluate the potential on-target side effects of interfering with the action of PICK1 action in CNS and beyond. We conclude that PICK1 constitutes a valid drug target for the treatment of inflammatory and neuropathic pain conditions without the side effects and abuse liability associated with current pain medication.

## 1. Introduction

### 1.1. Chronic Pain—An Enormous Burden to the Patients and Society

Acute pain in response to trauma or inflammation serves an important protective purpose conferring clear adaptive advantages. Chronic pain, on the other hand, has no protective purpose and does not confer any adaptive advantages and, as such, is considered maladaptive. Chronic pain may have both physical and psychological causes. Physical causes can involve musculoskeletal, vascular, neurological and oncological conditions, as well as injury to organs or originate from surgery. Chronic pain can be inflammatory, neuropathic (i.e., caused by neuronal injury) or a combination of both. Neuropathic pain, in particular, is complex and often resistant to commonly used medication. Chronic pain manifests as spontaneous pain (e.g., burning, shooting, and stabbing pain) as well as amplified evoked pain responses to noxious (hyperalgesia) or non-noxious (allodynia) stimuli.

Chronic pain is one of the ten most prevalent diseases worldwide affecting approximately 20% of the adult population, and the burden caused by chronic pain is escalating [1,2,3]. The prevalence, need for healthcare intervention and loss of productivity drive estimated direct and indirect healthcare costs to reach excessive amounts that surpass other major diseases, such as heart diseases, cancer and diabetes [4] On the individual level, chronic pain has a profound impact on the quality of life (QoL), often with detrimental effects on physical, psychological and social wellbeing. It can lead to reduced mobility and a consequent loss of strength, compromise the immune system, and interfere with a person’s ability to eat, concentrate, sleep, or interact with others. Current treatments, including opioids, anti-convulsants and anti-depressants are only moderately effective and limited by severe side effects, such as drowsiness, apathy, fatigue, loss of ability to function socially and professionally, which reduce QoL even further [3]. In addition, addiction resulting from pain treatment with opioids represents an urgent and immense problem with fatal consequences as opioid-related mortality is currently the main driver of drug overdose deaths in the U.S. (CDC/National Center for Health Statistics, 17 November 2021). Despite the obvious unmet medical need, development of analgesics with better efficacy and fewer side effects have been limited. Over the last decade, few new drugs have been marketed and most preclinical and clinical developments, according to our own GlobalData search, are centered around the same validated targets, including voltage and ligand gated ion channels, as well as GPCRs or targets related to inflammatory pain, including growth factors/cytokines and their related kinase-linked receptors.

### 1.2. Central Sensitization—A Key Component of Chronic Pain

Despite various etiologies of chronic pain, including the involvement of different neuropathic and inflammatory components, central sensitization, caused by increases in membrane excitability, synaptic efficacy, or a reduced inhibition, has emerged as a critical biological mechanism giving rise to convergence and amplification of nociceptive sensory inputs as well as the spontaneous sensation of pain [5]. At the molecular level, this maladaptive plasticity, which serves to amplify and sustain the sensation and perception of pain, is manifested by a myriad of events, including NMDA-type glutamate receptor (NMDAR)-dependent reorganization and insertion of excessive AMPARs in nociceptive synapses [5]. The AMPARs, composed of homo- or hetero-tetramers of the four basic units, GluA1–4, are highly mobile and dynamic, and therefore readily capable of changing the synaptic strength in response to changing inputs. Augmented AMPAR signaling appears both in inflammatory, post-operative, injury-induced and neuropathic pain models to be critical for development and maintenance of maladaptive synaptic strengthening of sensory signaling within the spinal cord [6,7,8,9,10]. It may even initiate an inflexible feedforward loop, where AMPAR complexes lacking the GluA2 subunit, known as calcium-permeable AMPA receptors (CP-AMPARs) [11,12], which allows frequent calcium entry, are upregulated to drive a vicious cycle that sustain aberrant AMPAR trafficking and signaling in the spinal cord, and hence invigorate central pain sensitization [13]. Central sensitization may even spread beyond spinal cord synapses and reach supraspinal areas [14,15,16,17]. Nevertheless, we lack supporting data showing how changes in AMPARs expression, composition, and signaling applies to human pain conditions, and how it may differ from rodent pain model. The efficacy of non-competitive AMPAR antagonist perampanel in human patients, however, is promising in this regard, despite the serious side-effects associated with this therapeutic approach. Nonetheless, the pharmacological interference of AMPAR signaling, including the use of a CP-AMPAR-specific antagonist, in model systems, strongly argues that therapeutic interventions addressing maladaptive AMPA receptor-mediated signaling in pain conditions could indeed constitute an attractive target for reliable pain control. For more elaborate details on AMPARs in pain biology, we refer to the recent review by Kopach and Voitenko [18], while further details of AMPAR pharmacology are reviewed by Chang, Verbich, and McKinney [19] and CP-AMPARs in disease conditions and their pharmacology are addressed by Cull-Candy and Farrant [20].

## 2. The Role of PICK1 in Chronic Pain

### 2.1. PICK1—A Central Regulator of AMPAR Trafficking and the Expression of CP-AMPARs

PICK1 (protein interacting with C kinase 1) is a functional dimer with two PDZ (PSD95/Disc-Large/ZO-1) domains flanking the central, membrane-binding BAR (Bin/amphiphysin/Rvs) domain that also mediates its dimerization [21]. As indicated by the name, PICK1 was identified as an interaction partner of protein kinase C [22,23]. PICK1 interacts via its PDZ domain also with the C terminus of several different membrane proteins, and it is believed that these interactions enable PICK1 to regulate the surface expression and phosphorylation states of the membrane proteins [24,25]. Indeed, PICK1 is part of regulatory network of proteins, including, e.g., SAP-97 (Synapse-Associated Protein 97) [26], GRIP (Glutamate Receptor Interacting Protein) [27] and NSF (N-ethylmaleimide-Sensitive Factor) [28] that all interact directly with the intracellular regions of AMPARs [29,30]. Based on the use of peptide inhibitors and mutational analysis, it was possible to establish a critical role of these proteins, including PICK1, in the synaptic stabilization of AMPARs through PKC-dependent phosphorylation, and thereby in synaptic plasticity, including, in particular, long-term depression (LTD) [31,32,33,34,35,36,37,38,39]. A model was proposed in which PICK1, as part of this regulatory network of proteins (and with the direct interaction with both GRIP1 [40] and NSF [41]), would scaffold PKCα at the C terminus of GluA2 to facilitate the phosphorylation of S880. In turn, this would dissociate AMPARs from GRIP1 and facilitate their internalization, perhaps with the direct assistance of PICK1, during LTD [37,42]. Subsequent studies, however, have complicated matters and suggested the direct involvement of PICK1 also in the stabilization of perisynaptic AMPARs [43], in regulation of AMPAR endoplasmic reticulum exit [44] and endosomal recycling [45]. The advent of the PICK1 knockout (KO) mouse confirmed the involvement of PICK1 in LTD [43] and some types of long-term potentiation (LTP) [46,47,48], although the details remain a matter of some controversy. The precise cellular and physiological function of PICK1 in activity-dependent plasticity appears, therefore, highly contextual, depending on the brain region, age, and stimulus. Finally, PICK1 is also involved in homeostatic scaling up, which is occluded in PICK1 KO neurons; however, it is not required for scaling down [49]. Of interest, despite this abundant literature on the involvement of PICK1 in synaptic plasticity, and most of the studies have been carried out on hippocampal tissue, the only PICK1-related memory defect reported, to our knowledge, is the impairment of inhibitory avoidance learning in adult PICK1 KO mice [48].

While it is well established that PICK1 can bind GluA2 containing, but not GluA2 lacking, AMPA receptors, parallel work has nonetheless suggested a role for PICK1 in regulating the surface expression of GluA2 lacking CP-AMPAR. The first evidence of such a role was obtained in hippocampal slices, demonstrating that viral overexpression of PICK1 increased CP-AMPAR expression [50]. This was followed by two studies showing that the inhibition or ablation of PICK1 in cerebellar stellate cells blocked the activity-dependent exchange of CP-AMPARs by GluA2 containing calcium-impermeable AMPARs [51,52]. Later, several studies implicated PICK1 in the transient insertion of CP-AMPARs during LTP both in hippocampus [53,54] and cortex during behavioral stimulation [55], but the functional importance of this transient insertion remains controversial. Nevertheless, in several pathological conditions giving rise to excess/abnormal neuronal stimulation, CP-AMPARs become stably integrated in the synapse and appear to significantly contribute to pathology. Such a stable insertion, in turn, is also PICK1 dependent. PICK1 was demonstrated to be involved in the cocaine-dependent insertion of CP-AMPAR in the ventral tegmental area (VTA) [56], as well as in the metabotropic glutamate receptor (mGluR)-dependent removal of these CP-AMPARs [57]. Likewise, CP-AMPAR insertion in the nucleus accumbens (NAc) after repeated cocaine self-administration was suggested to be PICK1 dependent [58,59]. Additionally, PICK1 was implicated in the pathological synaptic stabilization of CP-AMPARs following traumatic brain injury [60] and oxygen/glucose deprivation (OGD) leading to cell death [61].

Mechanistically, we suggest that PICK1 serves (at least) two molecular functions in relation to AMPARs through its PDZ domain interaction with the GluA2 C terminus (Figure 1). The first function involves an ability to localize and slow down the trafficking of GluA2 containing receptors both in the biosynthetic and recycling pathway as well as at the plasma membrane in the soma and dendritic shaft. These receptors constitute a pool of receptors, of which the surface receptors would be in relatively fast dynamic equilibrium with the synaptic receptors, whereas the intracellularly retained receptors would await activity-dependent phosphorylation for their mobilization. The second molecular function involves the ability to orchestrate this phosphorylation activity by the direct scaffolding of PKCα, CaMKII, and calcineurin, which, in turn, changes the dynamic properties of the newly inserted receptors giving rise to alterations in the synaptic integration and efficacy (Figure 1, left).

It might appear counterintuitive as to how this would affect the synaptic integration of CP-AMPARs, which do not include the GluA2 subunit and, consequently, are not directly associated with PICK1. There are several ways in which this could be envisioned, but the simplest is a phosphorylation-dependent destabilization of GluA2 containing surface receptors (e.g., on S880), thereby lowering their availability for synaptic integration by a mechanism similar to LTD. This would in turn favor the synaptic integration of GluA2-lacking CP-AMPARs, depending on, e.g., the GluA1 phosphorylation status of these receptors (Figure 1, right). This model obviously possesses a better explanatory than predictive power because of the overlapping kinetic parameters and unknown phosphorylation patterns for different receptor compositions, and it calls for modeling efforts to capture and dynamically integrate the existing data into a quantitative model with predictive power.

### 2.2. PICK1 as a Target in Pain

Soon after their discovery, the manner by which PDZ-domain scaffold proteins bind to the GluA2 C terminus affecting the spinal cord AMPAR transmission was investigated. The studies, which were based on the use of inhibitory peptides, pointed to a role of GRIP1 rather than of PICK1, with GRIP1 mediating the recruitment of AMPARs to silent synapses [62]. PICK1 was first implicated in pain pathology and AMPAR regulation in a study by Garry et al. in 2003 that probed the role of PICK1, GRIP1, and NSF in the hyperalgesic phenotype of rats subjected to the chronic constriction injury (CCI) model of neuropathic pain [6]. Cell permeable, myristoylated peptides of 11 residues mimicking the NSF and PDZ binding region of GluA2, respectively, were shown upon intrathecal administration to relieve thermal allodynia with a similar efficacy as AMPAR antagonists. The respective role of PICK1 and GRIP1 was further dissected using the S880 phospho-mimetic peptide, which was also effective in relieving thermal allodynia. Since this peptide does not bind GRIP1, PICK1 was suggested to be responsible for the increased GluA2 surface expression and central sensitization of the AMPAR transmission. Importantly, in all cases, the effect on mechanical allodynia was more modest. With the advent of the PICK KO mouse, the expression pattern of PICK1 in the PNS and spinal cord could be specifically addressed. Interestingly, Western blot analysis showed that PICK1 was expressed in both the dorsal and ventral horn of the spinal cord, as well as in the dorsal root ganglion (DRG) [63]. More detailed localization studies further confined PICK1 expression to small and medium peptidergic and non-peptidergic neurons in the DRG, and to lamina I and II in the dorsal horn with prominent pre- and postsynaptic localizations [64].

Another interesting question is the role of PICK1 in relation to the NMDAR-dependent phosphorylation of GluA2 S880 by PKC, and the subsequent internalization of GluA2 in inflammatory pain [65]. Indeed, the phosphorylation and membrane redistribution of GluA2 observed upon complete Freund’s adjuvant (CFA)-induced inflammation in WT mice were both abolished in PICK1 KO mice, confirming a role for PICK1 in the central sensitization paradigm. In accordance, both mechanical and thermal allodynia in response to CFA injection were shown to be attenuated in the PICK1 KO mice, while the normal response to thermal and mechanical stimuli was unaltered. Likewise, a PICK1 antisense oligodeoxynucleotide knockdown approach in rats demonstrated the attenuation of both mechanical and thermal allodynia after CFA injection. On the other hand, neither of the approaches affected the guarding behavior or thermal/mechanical allodynia in the incision-induced post-operative pain model, which fits with the notion that this model does not evoke GluA2 S880 phosphorylation and internalization or changes in GluA1 expression [63]. The same antisense knockdown methodology was subsequently used to address the role of PICK1 in the development of hyperalgesia in the spinal nerve ligation (SNL) model of neuropathic pain. In this model, no redistribution of AMPARs was observed, yet, again, the role of PICK1 in the pathology was confirmed as the PICK1 KO mice did not even develop allodynia in response to the mechanical or thermal stimuli, while the basal response to these stimuli was similar to WT mice [64]. Likewise, we recently showed that the intrathecal administration of a high affinity, Tat-conjugated bivalent inhibitor of the PICK1 PDZ domain, tPD5, fully relieved mechanical allodynia in the mouse spared nerve injury (SNI) model of neuropathic pain. The administration of tPD5 reduced GluA2 S880 phosphorylation in the spinal cord and, in the spinal cord slices from SNI animals, the peptide reversed the increased surface expression of GluA1 and GluA2 as well as the TNFα-induced insertion of CP-AMPARs [66]. In accordance with these findings, electrophysiological recordings in adult, decerebrate mice subjected to SNI displayed reduced field potentials in layers 1 and 2 of the dorsal horn upon the administration of tPD5. Somewhat surprisingly, field potentials were also reduced in the stimulated afferent nerve fibers of the spared sural nerve pointing to a putative function of PICK1 also in the peripheral nerves or surrounding tissue [66]. These observations could indicate that PICK1 inhibition may interfere with central sensitization by reducing both synaptic efficacy and membrane excitability. A peripheral function of PICK1 was further emphasized by our recent findings showing that the intraplantar injection of tPD5 alleviated mechanical allodynia in the CFA model of inflammatory pain [67], although this was not the case in the SNI model [66]. Finally, it was observed that electroacupuncture caused an increased clustering of PICK1 together with CamKII [68], as well as an upregulation in the spinal cord dorsal horn of Islet Cell Autoantigen 69 (ICA69) that forms heterodimers with PICK1 through its homologous membrane-binding BAR domain [69]. Furthermore, the anti-hyperalgesic effect of the electroacupuncture in the CFA model was impaired in ICA69 KO mice, suggesting that ICA69 can be regulated to counteract the PICK1 homomeric function in inflammatory pain transmission [69].

Collectively, these studies highlight PICK1 as an attractive target for the treatment of chronic pain conditions by substantiating the critical involvement of PICK1 in the development of neuropathic pain-related mechanical allodynia in particular. Mechanistically, it is plausible that this function of PICK1 is the result of the maladaptive regulation of AMPAR surface expression with the phosphorylation of GluA2 S880 as a reliable proxy. It remains to be shown, however, using, for example, the GluA2 K882A knock-in mice [43], whether this phosphorylation plays a causal role. In conclusion, we believe that the mechanism by which PICK1 regulates AMPAR surface expression, peripheral excitability, and function in pain deserves further attention.

### 2.3. Putative Additional Chronic Pain-Related Interaction Partners for PICK1

Despite the convincing evidence for the role of PICK1 in the regulation of AMPAR-dependent plasticity in chronic pain, it is worth considering that PICK1 interacts with many proteins in addition to PKCα and GluA2. Most of these interactions are mediated by the PDZ domain and, hence, all the manipulations to PICK1 mentioned above could be the result of blocking the binding of other interaction partners. To explore the overlap of the PICK1 interactome with the pain-related proteome, we took advantage of the bioinformatic tool StringApp [70] in Cytoscape (Figure 2a). This identified the heteromeric partner Islet Cell Antigen 69 (ICA69) (ICA1 and ICA1L) and a number of glutamate receptors, including the kainate receptor subunits (GRIK1 and 2) and the metabotropic receptors (GRK4 and 7), in addition to the AMPAR receptor subunits (GRIA2 and 3). Additionally, kinase PKB (AKT1) was identified in addition to PKCα. Additional channels include the Acid-Sensing Ion Channel, ASIC1 and 2, as well as Aquaporin 1 (AQP1). The monoamine transporters (SLC6A2, 3, and 4) stand out further, as does a series of kinase-coupled receptors. Pathway and functional analysis implicate PICK1 predominantly in the regulation of the dopaminergic synapse and AMPAR trafficking and function (Figure 2b). Taking advantage of the recently published mRNA expression analysis in mouse tissue [71], we here found confirmation of high expression of PICK1 in both the spinal cord and DRGs, compared to other tissues (Figure 2c). Importantly, the prominent expression of PICK1 is similarly observed in the tibial nerve, DRGs, and spinal cord in humans (Figure 2d), further consolidating PICK1 as a promising drug target [71]. The further analysis of human tissues by North et al. 2019, shows similar PICK1 expressions across sensory neurons and in male vs. females. Furthermore, the expression of PICK1 appears not to be regulated in pain conditions [72].

From the expression pattern of the pain-associated PICK1 interaction partners, we note that the heteromeric BAR domain interaction partner ICA69 is expressed in the DRG and spinal cord, and the relevant supraspinal regions. On the other hand, PKCα is not highly expressed in the DRGs or spinal cord, compared to the supraspinal areas in either mice or humans and, overall, the human expression is relatively low (Figure 2c,d). Notably, except for the expression of the AMPAR subunits in the spinal cord, a similar overall pattern is observed for the ion channels with which PICK1 interacts—but the peripheral and spinal cord expression of these channels, in particular, is relatively low in humans (Figure 2d). AQP1, on the other hand, is highly expressed in the DRGs in both mice and humans, attracting attention to its putative PICK1-dependent regulation in pain. Although, the overall expression of the kinase-linked receptors is comparatively low, they are indeed expressed in peripheral neurons, which also deserves further attention. Finally, although no expression of the monoamine transporters is observed in the spinal cord or other relevant tissues, it should be noted that the neurons expressing these transporters are localized to compact supraspinal nuclei; hence, it is likely that the mRNA is not detected in such analyses. Given that the noradrenalin transporter (NET) is known to be a potent target for pain-relieving drugs, such as amitriptyline [73], the role of PICK1 in the regulation of the NET surface expression in descending neurons should also be investigated.

## 3. Development of the PICK1 Inhibitors—Toward a Therapeutic Potential?

PICK1 is an untraditional intracellular target, in the sense that it is a scaffold protein without an enzymatic or independent signaling function. Moreover, the PDZ domains, in general, have proven to be challenging to target with sufficient selectivity and affinity. The first efforts to block the PICK1 PDZ domain employed a myristoylated version of the GluA2 C terminus, as previously described [6]; however, even by the intrathecal injection of a high concentration of this peptide, it was only modestly efficacious in alleviating mechanical allodynia in the SNL model. Additionally, we did not observe any effect on the mechanical allodynia after i.t. injection of this peptide in the SNI model in mice (unpublished observation). To develop small molecule inhibitors to PICK1, we screened a library of ~50,000 compounds and identified FSC231 as a reasonably selective inhibitor of PICK1, with an affinity of 10 µM [47]. However, the further improvement of the affinity and pharmacokinetic properties was unsuccessful [74]. FSC231 did not show any effect on the mechanical allodynia after i.t. injection in the SNI model in mice [66], but repeated administration for 7 days prior to paclitaxel injections prevented the development of hyperalgesia by a peripheral action involving a reduction in GSK-3β and ERK1/2 phosphorylation [75]. With a focus on PICK1 as a putative target in Alzheimer’s disease, Biogen also identified a series of PICK1 inhibitors, and although affinities down to 69 nM were obtained and the molecule was co-crystalized with the PICK1 PDZ domain (PDB: 6AR4), a further development was abandoned due to poor pharmacokinetic properties, as claimed by the authors [76,77]. These compounds, however, remain to be explored in relation to pain. Most recently, inspired by the development of high-affinity bivalent inhibitors of the PSD95 PDZ domains [78], we combined structural data from the best-known peptide ligand for the PICK1 PDZ domain mimicking the C terminus of the dopamine transporter (PDB: 2LUI) [79] with information on the oligomeric configuration of PICK1 demonstrated by small angle X-ray scattering (SAXS) (SASBDB: SASDAB8) [21], to develop tPD5 with an affinity of 1.7 nM [66]. We found that tPD5 effectively permeates cells and enters the spinal cord following i.t. injection, and even shows distinct, but not exclusive, neuronal tropism [66]. Additionally, tPD5 was shown to cross the blood–brain barrier to enter NAc, following injection into the jugular vein to attenuate cocaine seeking in rats [80]. Notably, tPD5 is highly similar to the Avilex Pharma compound AVLX-144, a Tat-conjugated bivalent inhibitor targeting PSD-95, for the treatment of ischemia following stroke, which recently passed phase 1 trials (ClinicalTrials.gov identifier: NCT04689035). This qualifies for operational feasibility in terms of up-scaling and the overall tolerance of bivalent Tat-conjugated peptides. Nonetheless, for chronic pain treatment and repeated administration, a strong drug candidate will have to demonstrate further improvement in stability, as well as the pharmacodynamic and pharmacokinetic properties.

## 4. Risk Assessment of PICK1 Inhibition

In this section, we evaluate the putative risks of PICK1 inhibition based on the available preclinical data from studies on PICK1 KO mice and from human genetic studies. Evidently, this picture could significantly diverge in humans, and the actual risk of a given compound would relate to biodistribution among other things. Nevertheless, our current knowledge would constitute the vantage point of a risk assessment to guide preclinical toxicology in a drug-development process.

### 4.1. Side-Effects Associated with Current Pain Drugs Are Unlikely for PICK1 Inhibitors

*General excitability*: current treatments for chronic pain, including anti-convulsants and opioids, are limited by their relative narrow therapeutic window between efficacious pain relief and side effects, such as drowsiness, apathy, and fatigue, which relate to their on-target actions on G_αi_-coupled opioid receptors and the voltage-gated calcium channel, respectively, to reduce synaptic transmissions throughout the CNS. Such side effects would not be anticipated by PICK1 inhibition from known mechanisms of action. Accordingly, PICK1 KO mice exhibit normal locomotor functions, including placing, grasping, and righting reflexes. Additionally, PICK1 KO mice do not display convulsions, hypermobility, spontaneous activity, or other significant difference in their general behaviors [38,63,64]. However, it is noteworthy that absence-like seizures, which are generalized nonconvulsive seizures characterized by a brief unresponsiveness to environmental stimuli and the cessation of activity, are induced more readily in PICK1 KO mice by carbamezine [81]. Likewise, i.v. administration of Tat peptides fused to peptides mimicking the C-terminal PDZ ligands from mGluR7 (TAT-R7) and GluA2 p880 (TAT-EVKI) induced absence-like seizures in mice and rats [81]. We have, however, not observed the behaviors corresponding to absence-like seizures upon the repeated administration of tPD5 [66,67].

*Compulsive behavior*: addictive side effects of pain-relieving drugs, and, in particular, for opioids, is another key concern. For novel drug targets for chronic pain, it is thus of outmost importance that they do not lead to reinforcing behaviors. In PICK KO mice, we showed that the acute locomotor response to a single injection of cocaine was markedly attenuated. Moreover, in support of the role of PICK1 in neuroadaptive changes induced by cocaine, we observed a diminished self-administration of cocaine in the KO mice [82], and the conditional deletion of PICK1 in the medial prefrontal cortex attenuated cue-induced cocaine seeking in male mice [83]. In accordance with the results, i.v. administration of tPD5 was shown to reduce the reinstatement of cocaine seeking in rats [80]. This suggest that PICK1 does not constitute a neutral target with respect to reinforcing behavior, but its inhibition might actually counteract the reinforcing effects of, for example, opioids, although the gender aspects in relation to abuse liability need a more thorough investigation [83]. Consequently, pharmacological interference with the PICK1 function could serve as an effective substitution/transition treatment option for opioids, for example, in late stages of post-surgical pain relief, and for chronic pain patients suffering from opioid addiction.

### 4.2. Putative CNS-Related Side Effects

*Memory function*: given the wealth of data involving PICK1 in synaptic plasticity, perhaps the most immediate concern for PICK1 as a drug target is centered around memory performance and flexibility. Nonetheless, the only finding implicating PICK1 in memory function in vivo is a reduced latency in the step-through inhibitory avoidance test observed in adults, but not juvenile PICK1 KO mice [48]. Since this test also involved a foot shock to induce the aversive behavior and relied on motivational aspects in addition to memory, the exact nature of this deficit was difficult to pinpoint. In the Barnes Maze, the i.p. administration of tPD5 during the training sessions did not compromise the learning of the task, as well as the long-term memory retention or reversal learning [67]. Although this was promising in relation to PICK1 as a drug target, further experiments are warranted to rule out the idea that interfering with the function of PICK1 in synaptic plasticity leads to unwanted on-target side effects.

*Alzheimer’s Disease*: a genetic analysis addressing the association between the PICK1 gene and the risk of Alzheimer’s disease (AD) in the Chinese Han population identified 7 single nucleotide polymorphisms in the PICK1 gene, with rs149474436 being less and rs397780637 being more frequent in AD patients [84]. This study was based on investigation of a small cohort and without further genetic information about the cohorts, meaning that the studies do not fulfill the current standards for candidate-gene identification, and PICK1 has not been identified as a risk gene in an AD Genome-Wide Association Study (GWAS) meta-analysis [85]. On the other hand, both ICA69 and ICA1L, which appear to antagonize the PICK1 function, were identified among 42 new risk loci for AD [86], suggesting that the inhibition of PICK1 might be protective against AD. In line with this notion, the synapto-depressive effects of amyloid-β were compromised in PICK1 KO, as well as by the small molecule inhibitor BIO922 [77,87].

*Schizophrenia and cognition*: some studies proposed PICK1 as a schizophrenia susceptibility gene. This association is partly coupled to the PICK1 PDZ domain interaction with serine racemase, an enzyme that converts L-serine to D-serine in glial cells [88]. Serving as an endogenous co-agonist for the glycine site of the NMDA receptor, D-serine, and hence serine racemase activity, is critical for the regulation of glutamatergic neurotransmission, and evidence has implicated the hypofunctioning of NMDA receptors in schizophrenia. The connection between PICK1, serine racemase, and NMDA hypofunction was highlighted, since juvenile PICK1 KO mice were shown to display lower D-serine levels in the prefrontal cortex and hippocampus. No differences were, however, detected in adult PICK KO mice [89]. In addition, some polymorphisms in the PICK1 gene (rs3952, rs713729, and rs2076369) have been linked to schizophrenia [90,91], although disputed [92]. It has also been suggested that the rs2076369 polymorphism is linked to cognition [93], but similar to the other investigations, this study was based on a small cohort investigating only two polymorphisms in the PICK1 gene. No other genetic information about the cohorts is available, meaning that the studies do not fulfill the current standards for candidate-gene identification (better knowledge about the genetic background of the participants is a prerequisite for any reliable conclusion). Moreover, the data from the latter study are somewhat conflicting, as there was no consistent correlation between homozygosity and heterozygosity for the polymorphism that was suggested to be relevant for cognitive functions. It is also important to mention that the PICK1 gene was not among the 108 schizophrenia loci identified by a genome-wide association study of up to 36,989 schizophrenic cases and 113,075 controls [94]. Ultimately, given the weak experimental support, we do not consider the suggested poorly documented link of PICK1 to cognitive and behavioral phenotypes in humans to be an important concern.

### 4.3. Putative Peripheral Side Effects

*Male infertility*: PICK1 homozygous KO males are infertile, whereas heterozygotes demonstrate fertility comparable to wild-type mice [95]. Male fertility was rescued following lentiviral PICK1′s expression in the testis by seminiferous tubule microinjection [96]. Moreover, a homozygous missense mutation (G198) was identified in a candidate-gene screening study of patients suffering from globozoospermia [97]. Mechanistically, PICK1 was shown to interact with Golgi-associated PDZ- and the coiled-coil motif-containing protein (GOPC) and the primary catalytic subunit of casein kinase 2 (CK2α′), proteins of which the deficiencies lead to globozoospermia in mice. PICK1 colocalized with GOPC in the Golgi apparatus and the absence of PICK1 lead to the fragmentation of acrosome formation, suggesting a function of PICK1 vesicle trafficking from the Golgi apparatus to the acrosome in the early spermatogenesis [95]. Taken together, this data implicates PICK1 in male infertility, whereas a phenotype for female mice has not been reported. However, the role that the PICK1 PDZ domain plays in relation to the Golgi to acrosome vesicle trafficking is unclear, with neither GOPC nor CK2α′ having a PDZ-binding motif at their respective C termini. Nevertheless, PICK1 inhibitors should be tested for their effect on male fertility.

*Growth retardation and glucose homeostasis*: the role of PICK1 in metabolic regulation was first implied by the distinct expression in peptidergic neurons in fruit flies [98], and in support of such a function, PICK1 knockout mice display growth retardation and impaired glucose homeostasis [99,100]. Four coding mutations in PICK1 were identified in a cohort of 1000 diabetic patients, suggesting a similar function in humans [101]. These deficits involve the impaired secretion of growth hormone and insulin, respectively, and relate to the role of PICK1 in the biogenesis of dense core vesicles (DCVs) [99,100] and maturation [102]. Similar to the role in acrosome formation, PICK1 was suggested to function in budding from the trans-Golgi network during DCV biogenesis [99], and it is unclear whether this function involves the PDZ domain. Functional characterization, however, demonstrated that the function relies on an intact BAR domain [103]. Recently, a role for PICK1 in the regulation of small synaptic vesicles during a high-frequency stimulation was also identified, but the physiological importance of this role remains to be determined [104].

*Cancer*: several studies have observed altered PICK1 expression in human cancers, but the results have been somewhat conflicting [105,106,107,108,109]. A study on breast, lung, gastric, colorectal, and ovarian cancer cells showed the upregulation of PICK1 mRNA and protein, when compared to adjacent normal epithelia, and that the upregulation correlated with a shortened span of overall patient survival [105]. Additionally, the siRNA-mediated knock down of PICK1 in MDA-MB-231 cells decreased cell proliferation and colony formation in vitro and inhibited tumorigenicity in nude mice [105], presenting PICK1 as a putative target in cancer. More recently, a transcriptome-wide Mendelian randomization study implicated an increased PICK1 expression as the most prominent risk factor for glioma susceptibility [110]. On the other hand, the effect of the microRNA, miR-615-3p, in promoting the epithelial–mesenchymal transition (EMT) in breast cancer cells was shown to rely on targeting the 3′-untranslated regions of PICK1 causing the inhibition of PICK1 translation and an increase in the downstream signaling of transforming growth factor β (TGFβ) [108]. Indeed, PICK1 was shown to decrease TGFβ signaling by the caveolin-dependent degradation of the TGFβ type I receptor and downstream phosho-Smad2 in a PDZ domain-dependent manner [111]. Other studies revealed reduced levels of PICK1 in gastric cancers, likewise increasing TGFβ signaling and EMT [109,112]. Finally, the reduced expression of PICK1 was observed in grade IV astrocytic tumor cell lines and grade IV tumors progressed from lower-grade tumors. The exogenous expression of PICK1 in the grade IV astrocytic cell line, U251, reduced its capacity for anchorage-independent growth, two-dimensional migration, and invasion through a three-dimensional matrix [106]. Taken together, these correlative and functional data suggest that PICK1 may also play a protective role in preventing cancer progression, and consequently PICK1 inhibitors should be tested for their effect on TGFβ signaling and EMT.

## 5. Conclusions

The research conducted over the last two decades has implicated PICK1 in the maladaptive plasticity of AMPAR, which is critical to central sensitization in chronic pain conditions; however, in the present study, we stress that several other PICK1 interaction partners are implicated in pain. The role of PICK1 in the condition of chronic pain was demonstrated by the use of membrane-permeable PDZ-binding peptides, KO mice, and antisense knock-down strategies; however, the development of small molecules to inhibit the PICK1 PDZ has proven to be a challenge. The recent advent of a high-affinity bivalent PDZ-domain peptide inhibitor may pave the way to the development of a novel efficacious treatment of chronic pain conditions. Current data and analysis of the putative on-target side-effect profile suggest little overlap with the side-effect profile of current pain medication involving addiction, drowsiness, apathy, and fatigue, but warrants a further investigation of the putative effects on memory and cognition, as well as on male fertility and neoplasm. In summary, we believe that PICK1 stands out as a highly promising novel drug target to address the significant unmet need of chronic-pain patients.

## 6. Materials and Methods

For Figure 1, pain-associated genes were extracted from five databases: the Human Pain Genetics Database (HPGD) [113], the Pain Genes Database (PGD) [114], GeneCards^®^ filtered using the keyword “Pain” and restricted to genes with a relevance score higher than 5, as well as the String Database (String-db) using the disease query “Pain disorder” [115]. All databases were accessed on the 13 October 2021. Protein-coding genes were extracted from all databases, resulting in 1138 unique genes. Subsequently, all Homo sapien (taxid:9606) genes were mapped to their respective UniProtKB accession identifiers (UniProtKB AC/ID) (https://www.uniprot.org, accessed on 13 October 2021), resulting in a database containing 1122 proteins. The UniProtKB AC/IDs were used to build a functional protein-association network with the stringApp [70] in Cytoscape (version 3.8.2, java 11.0.6) as well as the following apps: yFiles Layout Algorithms and enhancedGraphics. Using stringApp, we also imported a network for PICK1 (UniProtKB AC/ID: Q9NRD5) Homo sapiens (taxid:9606), with a confidence score cutoff of 0.3, resulting in 297 interaction partners. A sub-network was extracted containing the undirected first neighbors of PICK1 with a String-db experiment score, resulting in 105 interactors. The functional protein-association networks for pain and PICK1 were subsequently merged, based on the network intersection, resulting in 39 pain-associated interactors. The UniProtKD AC/IDs were exported to a .csv file and re-imported using the stringApp to identify all existing edges in the String-db. Finally, the list of resulting interactors was manually refined by investigating the literature, resulting in 22 interactors. The functional enrichment analysis for the 22 PICK1 interactors in Figure 1 was performed using the stringApp with the genome as the background. The false discovery rate was controlled by correcting the Fisher’s exact test *p*-values using the Benjamini–Hochberg procedure (the adjusted *p*-values are referred to as FDR). The FDR threshold was set to 0.01 and terms were filtered for redundancy with a cutoff of 0.5 (default setting in the stringApp), resulting in 26 significant terms from the GO Biological Process, GO Molecular Function, GO Cellular Component, and KEGG Pathways. The terms were manually curated to exclude broad terms and terms with less than 3 associated genes, resulting in 16 terms.

## 7. Patents

Technologies describing tPD5 and related work are disclosed in two patent applications presently being handled by the patent authorities in the E.U. and U.S.A.

## Figures and Tables

**Figure 1 cells-11-01255-f001:**
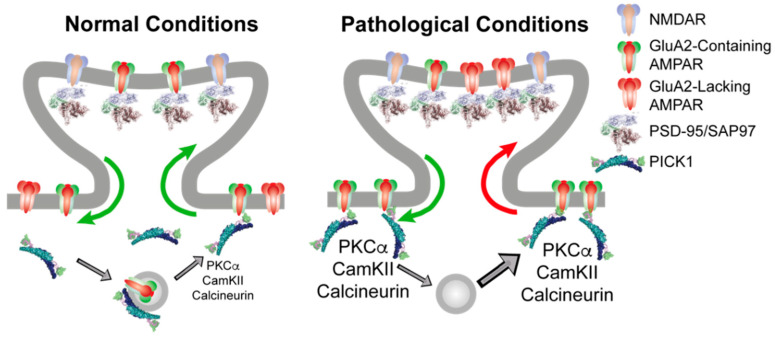
PICK1 serves an indirect role in regulating the synaptic AMPAR level and composition. (**Left**) Under basal conditions, PICK1 can anchor GluA2 containing AMPARs at extrasynaptic sites, as well as in intracellular vesicle stores, either derived from the endoplasmatic reticulum or belonging to recycling compartments. Synaptic and extrasynaptic GluA2-containing receptors are in continuous exchange, as indicated by green arrows, depending on the phosphorylation pattern, which, in turn, is subject to regulation by PICK1. (**Right**) Following repeated high-intensity stimulation giving rise to NMDAR activation and Ca^2+^ influx, there is an increase in the total AMPAR surface expression from PICK1 containing intracellular compartments. This increases the kinase activity to modify AMPAR C-terminal phosphorylation patterns in a PICK1-dependent manner, leading to increased PICK1-dependent stabilization of GluA2 containing receptors at extrasynaptic sites (green arrow). This extrasynaptic trapping of GluA2-containing AMPARs allows GluA2-lacking CP-AMPARs to access the synapse, giving rise to increased Ca^2+^ influx and an inflexible synaptic strengthening.

**Figure 2 cells-11-01255-f002:**
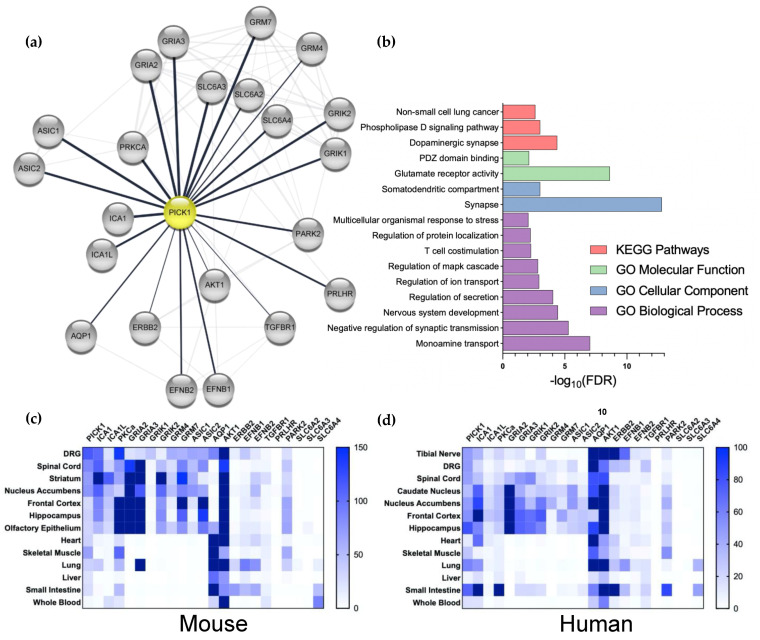
Analysis of the interaction and expression patterns of PICK1. (**a**) The stringApp (Doncheva et al. 2019) was used in Cytoscape to merge the network of pain-related proteins (1122 proteins from 5 databases, see Material and Methods) with PICK1 interaction partners (105 proteins, determined by the experiment score) to yield a functional protein-association network for Pain and PICK1 with 39 proteins. This network was manually validated by a literature search to yield a PICK1 network of 22 interactors. (**b**) The network was analyzed with respect to the biological pathways, processes, functions, and cellular compartments; FDR; Benjamini–Hochberg-adjusted *p*-values (see Section 6). (**c**,**d**) heatmap of the mRNA expression of PICK1 and its pain-related interaction partners in selected mice (**c**) and humans (**d**) Tissue. Dorsal root ganglion (DRG), spinal cord, and tibia nerve (human) are the primary pain-related tissues.

## Data Availability

Not applicable.

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
