# Peer review of "The Scaffold Protein PICK1 as a Target in Chronic Pain"

_cells, 2022, doi:10.3390/cells11081255_

Round 1

Reviewer 1 Report

This review of Toft Sorensen et al., evaluate the role of PICK1 considering this as a new potential target for chronic pain treatment. In particular the authors have examination the regulation of AMPARs by PICK1 and its relation to pain pathology, have presented several potential new PICK1 inhibitors and finally they have listed a possibly side-effect of these drugs. The article is well writing, the topic is very interesting, it is appreciating the effort of the authors in trying to simplify a complex argument, such as the transmission inherent in the activation/inhibition of the AMPA receptor and all the series of regulations that characterizes it.

I have few revisions.

MAJOR REVISIONS

  • I personally find it uninteresting and I think it is also risky to evaluate the possible side-effects of PICK1 inhibitors, the authors hypothesize and list what may or may not (for example the confront with opioids) happen with the use of these compounds. This series of toxic manifestations are based solely on preliminary studies that have not been evaluated and confirmed in humans, furthermore, considering that there are strong controversies on the possible mechanism of action of these drugs, I can hypothesize that even on the side-effects there could be some disputes.

              I therefore suggest that the title of chapter 4 and the paragraphs be changed in order to avoid any criticism.

  • It would have been interesting to evaluate and underline the pros and cons of PICK1 inhibitors vs other AMPA antagonists, such as Selective and non-competitive AMPA antagonists, potent and selective non-NMDA iGluR antagonists, voltage- and use-dependent open-channel AMPA blockers, Ca2 + -permeable AMPA antagonists, GluA1 and GluA3 selective AMPA antagonists. Pharmacokinetic characteristics, routes of administration, single and / or repeated treatment, activity in acute pain and chronic pain, possible other uses etc. The authors could add a chapter or to comment these aspects and highlight the real advantage of using the PICK1 inhibitors.

MINOR REVISIONS

  • I suggest to insert a small figure on the two molecular functions hypothesized by authors (at the end of paragraph 2.1, line 143), this could facilitate the understanding to the reader.

Author Response

MAJOR REVISIONS

  • I personally find it uninteresting and I think it is also risky to evaluate the possible side-effects of PICK1 inhibitors, the authors hypothesize and list what may or may not (for example the confront with opioids) happen with the use of these compounds. This series of toxic manifestations are based solely on preliminary studies that have not been evaluated and confirmed in humans, furthermore, considering that there are strong controversies on the possible mechanism of action of these drugs, I can hypothesize that even on the side-effects there could be some disputes. I therefore suggest that the title of chapter 4 and the paragraphs be changed in order to avoid any criticism.

We have changed the title of chapter 4 and written a brief introduction to the chapter to motivate it and prevent misunderstandings, so it now reads

‘4. Risk Assessment of PICK1 inhibition

In this section, we evaluate the putative risks of PICK1 inhibition based on available preclinical data from studies on the PICK1 KO mice and from human genetic studies. Obviously, this picture could diverge significantly in human and actual risk of a given compound would relate to biodistribution among other things. Nevertheless, our current knowledge, would constitute the vantage point of a risk assessment to guide preclinical toxicology in a drug development process.’

We are not aware of the ‘strong controversies on the possible mechanism of action of these drugs’ referred to. Also, it is not clear what drugs are referred to exactly.

  • It would have been interesting to evaluate and underline the pros and cons of PICK1 inhibitors vs other AMPA antagonists, such as Selective and non-competitive AMPA antagonists, potent and selective non-NMDA iGluR antagonists, voltage- and use-dependent open-channel AMPA blockers, Ca2 + -permeable AMPA antagonists, GluA1 and GluA3 selective AMPA antagonists. Pharmacokinetic characteristics, routes of administration, single and / or repeated treatment, activity in acute pain and chronic pain, possible other uses etc. The authors could add a chapter or to comment these aspects and highlight the real advantage of using the PICK1 inhibitors.

We believe a one-by-one comparison to the numerous AMPAR antagonists would be a very extensive effort and clearly outside our field of expertise. Further, given that the focus of the special issue is ‘Update on Molecular Mechanisms and Potential Drug Targets in Chronic Pain’, we believe that an extensive review of AMPAR antagonists and their pharmacology is outside the scope of the review. We have included, however, an additional review reference (new ref 19) on the topic at the end of the introduction (Chapter 1). This sentence now reads:

’For more elaborate details on AMPARs in pain biology, we refer to recent review by Kopach and Voitenko [18], while further details on AMPAR pharmacology is reviewed by Chang,  Verbich, and McKinney [19] and CP-AMPARs in disease conditions and their pharmacology is addressed by Cull-Candy and Farrant [20].’

MINOR REVISIONS

  • I suggest to insert a small figure on the two molecular functions hypothesized by authors (at the end of paragraph 2.1, line 143), this could facilitate the understanding to the reader.

We have included a figure (new figure 1) and legend to describe our suggested function for PICK1 in regulation of synaptic localization of AMPARs as suggested. Old figure 1 is changed to figure 2 accordingly

Reviewer 2 Report

After carefully reading the manuscript, I conclude that the abstract, introduction, and other chapters cover the issues discussed in an extensive and proper manner.

Although it is a valuable work having an interesting idea it needs some adjustment:

  • Considering the fact that the authors consider the PICK1 scaffold protein as a target in chronic pain, it seems advisable to discuss the available 3D models such as the PICK1-PDZ (RCSB PDB: 2LUI) used in the docking process. I also suggest to use for this purpose databases such as RCSB PDB (https://www.rcsb.org) or UniProtKB (https://www.uniprot.org/) or another. I propose at least to provide specific identifiers based on chosen database. Please complete the manuscript with these data in the form of a separate paragraph.
  • Since a large number of abbreviations are used along the text, I recommend that all used abbreviations should be listed at the end of the main body of text before references. This will certainly provide the reader with a better understanding of the context of the issues discussed. In addition, this is a kind of standard in contemporary scientific papers.
  • The small number of citations of works from the last 5 (after 2017) years is rather surprising (only 29 works out of 110 references have been published after 2017). I believe that the topic has been developing vigorously over the past 5 years and focusing on old articles is not the best approach to reviewing it. Therefore, I am convinced that the authors must carry out a detailed literature review and supplement the citations with references to the latest research in this field.

I recommend publication after minor revision.

Author Response

Comments and Suggestions for Authors

  • Considering the fact that the authors consider the PICK1 scaffold protein as a target in chronic pain, it seems advisable to discuss the available 3D models such as the PICK1-PDZ (RCSB PDB: 2LUI) used in the docking process. I also suggest to use for this purpose databases such as RCSB PDB (https://www.rcsb.org) or UniProtKB (https://www.uniprot.org/) or another.I propose at least to provide specific identifiers based on chosen database. Please complete the manuscript with these data in the form of a separate paragraph.

Thank you for pointing out this omission. We have now included reference to PDB: 6AR4 for the biogen small molecule co-crystalized with the PICK1 PDZ domain (line 298) as well as PDB: 2LUI for the DAT C-terminal interaction with the PDZ domain (line 302). Also, we have included a reference to the Small Angle X-Ray database SASBDB: SASDAB8 for the oligomeric small angle scattering structure of PICK1.

  • Since a large number of abbreviations are used along the text, I recommend that all used abbreviations should be listed at the end of the main body of text before references. This will certainly provide the reader with a better understanding of the context of the issues discussed. In addition, this is a kind of standard in contemporary scientific papers.

We have included a list of abbreviations as recommended

  • The small number of citations of works from the last 5 (after 2017) years is rather surprising (only 29 works out of 110 references have been published after 2017). I believe that the topic has been developing vigorously over the past 5 years and focusing on old articles is not the best approach to reviewing it. Therefore, I am convinced that the authors must carry out a detailed literature review and supplement the citations with references to the latest research in this field.

We have indeed carried out a detailed literature review and have included all relevant data related to the topic, including publications since 2017 without bias (29 of 110 (0.26) references after 2017 actually gives a slight preference towards newer papers than the overall fraction off papers referring to e.g. PICK published after 2017 (76 of a total of 367 (0.2)). Nonetheless, going over the literature again, we have chosen to include 3 additional studies from 2021 that have been published during writing of the review. These are new ref 68 (integrated at line 228), new ref 83 (integrated at line 355 and 360) and new ref 110 (integrated at line 458). We would be happy to include specific references as suggested by reviewers/editor that we may have overlooked.